# Education Practices of Dietitians Across Australia and New Zealand Around the Glycaemic Management of Dietary Fat and Protein in Type 1 Diabetes and the Use of Continuous Glucose Monitoring: A Survey Evaluation

**DOI:** 10.3390/nu17071109

**Published:** 2025-03-22

**Authors:** Evangeline Laurence, Carmel E. Smart, Kirrilly M. Pursey, Tenele A. Smith

**Affiliations:** 1College of Health, Medicine, and Wellbeing, University of Newcastle, Callaghan, Newcastle, NSW 2308, Australia; evangeline.laurence@health.nsw.gov.au (E.L.); kirrilly.pursey@newcastle.edu.au (K.M.P.); tenele.smith@newcastle.edu.au (T.A.S.); 2Mothers and Babies Research Centre, Hunter Medical Research Institute, New Lambton Heights, Newcastle, NSW 2305, Australia; 3Department of Paediatric Endocrinology, John Hunter Children’s Hospital, New Lambton Heights, Newcastle, NSW 2305, Australia; 4Food and Nutrition Research Program, Hunter Medical Research Institute, New Lambton Heights, Newcastle, NSW 2305, Australia

**Keywords:** diabetes mellitus, type 1, dietary fats, child, continuous glucose monitoring, insulin

## Abstract

**Background/Objectives**: International guidelines recommend that all children and adolescents with type 1 diabetes (T1D) receive education on the glycaemic impact of fat and protein from diagnosis. In addition, the insulin strategy should be adjusted to compensate for fat and protein excursions. Data from continuous glucose monitoring (CGM) can guide insulin adjustment. This study sought to determine whether the current practices of dietitians in Australia and New Zealand align with guidelines. **Methods**: An anonymous, online survey of paediatric T1D dietitians working in tertiary centres (n = 20; Australia, n = 14, New Zealand, n = 6) was undertaken from February to March 2023. The Australian and New Zealand Society for Paediatric Endocrinology and Diabetes (ANZSPED) disseminated the survey link. The questionnaire covered three content domains: demographic information about the clinic and practitioner, the health professionals’ education practices regarding fat and protein, and the use of CGM. **Results**: This pilot study had a 100% response rate, with a dietitian representative from all eligible centres responding on behalf of the diabetes team. Only 10% (n = 2) of respondents both (i) provided education on the glycaemic impact of fat and protein to all families at diagnosis and (ii) always provided insulin strategies to manage fat and protein where it impacted glycemia, as per guidelines. Barriers to education included a lack of procedure (47%, n = 7), consumer resources (40%, n = 6), and time (33%, n = 5). Reasons for not recommending strategies to manage fat and protein were perceptions that the family was overwhelmed (100%, n = 10) or not interested (60%, n = 6), and uncertainty of the best strategy (40%, n = 4). CGM was used by “almost all” respondents to educate and adjust the insulin strategy (90%, n = 18). **Conclusions**: Most dietitians surveyed were not consistently providing fat and protein education and management strategies to children with T1D in line with guidelines. CGM is a key tool routinely used by dietitians in nutrition education to help guide insulin adjustment. Dietitians need greater support through educational resources for families and training in evidence-based strategies to manage deglycation from dietary fat and protein to align with guidelines.

## 1. Introduction

In people living with type 1 diabetes (T1D), postprandial glycaemia has been identified as the single largest factor contributing to HbA1c and, therefore, a critically important therapeutic target for the prevention of diabetes-related vascular complications [1]. There are more than 15,000 children and young people (0–19 years) across New Zealand [2,3] and Australia living with T1D, more than three-quarters of whom do not meet recommended HbA1c targets [4]. This is despite the introduction of technologies such as automated insulin delivery (AID) systems that have been shown to help meet glycaemic targets [5].

To optimise postprandial glycaemia and minimise complication risk, medical nutrition therapy (MNT), delivered by a specialist paediatric T1D dietitian is recommended from diagnosis [6]. MNT has long been the cornerstone of T1D management and promotes a healthful, varied diet that supports growth and development with a method of adjusting insulin to carbohydrate intake. However, over the past decade, research has demonstrated that fat and protein, independently [7,8,9] and combined [10], modulate the response to carbohydrate, delaying the early postprandial glucose rise by up to 3 h and resulting in sustained hyperglycaemia that can persist for up to 12 h. The magnitude of this excursion has been shown to vary significantly between individuals despite the same meal content, necessitating an individualized approach to insulin dosing for fat and protein [11].

To achieve target postprandial glycaemia, best-practice paediatric T1D nutrition guidelines now recommend that, in addition to carbohydrate, people with T1D receive education on the glycaemic impact of fat and protein at diagnosis and adjust the insulin dose and delivery to compensate for their delayed and prolonged glycaemic impact [6,12]. Further, CGM has been identified as a key tool that can be used to guide individual dose adjustment [6].

Our research has shown that families living with T1D are willing to adopt strategies beyond carbohydrate counting to optimise postprandial glycaemia and identified health-care professionals as the primary source of information on insulin strategies to manage meals [13]. Despite this, no studies have examined if guidelines on the management of dietary fat and protein have been implemented by health care professionals in clinical practice. In addition, no studies have explored the use of CGM as a tool in facilitating education and guiding adjustments to the insulin strategy for fat and protein meals. It is important to understand current practices to tailor future education for health professionals regarding fat and protein management and the use of CGM in individualizing dietary and insulin dosing advice for children and adolescents with T1D.

The primary aim of this study was to evaluate whether the current practices of dietitians in Australia and New Zealand are in line with international T1D nutrition guidelines. A secondary aim was to investigate if dietitians are using CGM as a tool to assist in interventions to manage fat- and protein-related glycaemic excursions.

## 2. Materials and Methods

### 2.1. Study Design and Participants

In this pilot study, dietitians in Australia and New Zealand caring for children and adolescents (0–17.9 years) living with T1D completed an anonymous, online survey between February and March of 2023. Dietitians were eligible to participate in the survey on behalf of the diabetes team if they were currently employed in Australia or New Zealand at a tertiary centre caring for ≥100 children and adolescents with T1D aged 0–17.9 years. Dietitians working at tertiary centres were selected to complete the survey as (i) dietitians undertake the majority of nutrition education provision in diabetes care [6] and (ii) tertiary centres generally provide outreach to rural and remote centres and are often responsible for supporting education of other health professionals state-wide. Ethics was approved by University of Newcastle Human Ethics Advisory Panel (approval number: H-2023-0036).

### 2.2. Survey Design

The survey was developed by paediatric T1D dietitians, who were not involved in survey completion, and a diabetes nutrition researcher. The questions in the survey were modelled on queries put forth by diabetes teams at national and international meetings regarding the glycaemic impact of fat and protein and insulin dosing for fat and protein consumption. This provided direction for survey content and insight into potential areas where implementation of teaching in clinical practice could be challenging. The questionnaire covered three content domains: demographic information about the clinic and practitioner, the health professionals’ education practices regarding fat and protein, and the use of CGM. Data on the respondents’ confidence in fat and protein education and CGM use were collected as part of the second two domains (See Appendix A). The final survey consisted of between 32–45 questions (multiple choice, multiple select, or short response) and took 15–20 min to complete. This study aligned with The Checklist for Reporting Results of Internet e-Surveys (CHERRIES) [14].

The survey was pilot-tested by two specialist paediatric diabetes dietitians not currently working in the area, one endocrinologist, and one credentialed diabetes educator. They were asked to comment on the clarity and appropriateness of questions and terminology and the logic of the question ordering. Subsequently, minor changes were made to the wording of questions to optimise clarity and flow.

### 2.3. Survey Dissemination

The Australia and New Zealand Society for Paediatric Endocrinology and Diabetes (ANZSPED) disseminated the survey. ANZSPED represents health care professionals working in Australia and New Zealand and promotes effective care for children living with diabetes and other endocrine disorders [15]. An invitation to complete the survey and survey link were sent to the heads of tertiary departments via email, with a request to forward to their centres’ dietitian for completion within 4 weeks. The invitation stipulated that the survey be completed by only one eligible dietitian from each centre, the most senior dietitian where there was more than one. Representation from each state in Australia and all major centres in New Zealand was sought.

### 2.4. Data Collection

The survey responses were collected online via a web-based electronic data capture, management, and survey tool, REDCap [16,17]. There was no limit to the number of times the survey could be exited and returned to; however, every question required a response before the participant could progress. Participants were not able to backtrack, and randomisation of questions was not possible due to the branching logic of the survey. The link was unable to be used after completion of the survey, thereby ensuring participants were not able to complete the survey multiple times.

### 2.5. Data Analysis

Survey data were analyzed descriptively, with categorical data represented by frequency (%) and continuous or discrete data presented as a median and range. Associations between categorical data sets were analysed using a Fisher’s Exact test due to small cell counts, where a *p* value of less than 0.05 was considered significant. Short-answer questions were analysed for common themes by two members of the research team and then triangulated by a third member. The term ‘respondent’ refers to the individual dietitian completing the survey.

## 3. Results

Twenty eligible dietitians representing 20 different tertiary paediatric T1D centres across Australia (70%, n = 14) and New Zealand (30%, n = 6) were invited and completed the survey (response rate of 100%).

### 3.1. Respondent Characteristics

Across respondents, 40% (n = 8) had between 6–10 years of experience (mean 11 ± 6.3 years) working with children and adolescents with T1D. Respondents worked at centres located in metropolitan (70%, n = 14) or regional (30%, n = 6) areas that cared for a median of 200 children and adolescents with T1D, with a range of 70–1500 (one clinic in the Northern Territory of Australia with <100 children was included, as it was deemed important to represent all states and territories). Most respondents reviewed children and adolescents once (n = 9, 45%) or twice (n = 8, 40%) per year and taught carbohydrate counting from diagnosis (90%, n = 18). Further demographic information can be found in Table 1.

### 3.2. Prevalence of Care Addressing the Glycaemic Impact of Fat and Protein and Management Strategies

One quarter of respondents (n = 5) provided education on the glycaemic impact of fat and protein to all families (Figure 1), and half (50%, n = 10) always provided strategies to manage fat and protein where there was evidence of an impact on postprandial glycaemia (Figure 1). Only 10% (n = 2) of all respondents provided education to all families at diagnosis and always provided strategies to manage fat and protein where there was evidence of an impact, consistent with best-practice international nutrition guidelines.

### 3.3. Methods of Fat and Protein Education

All respondents believed that providing education on the glycaemic impact of fat and protein and management strategies were the dietitian’s role, yet none of the respondents had a policy or procedure in place at their centre that detailed what information should be provided, when, to whom, and how it should be provided. The most common timepoint for the delivery of fat and protein education was within the first year of diagnosis (45%, n = 9), followed by when the clinician noticed problematic glycaemic excursions (15%, n = 3), when the family noticed problematic glycaemic excursions (15%, n = 3) and at diagnosis (15%, n = 3) (Figure 2).

Of those respondents who did not provide education on the glycaemic impact of fat and protein to all families (70%, n = 14), the majority (71%, n = 10) provided education to most families (classified as >50% of families), while only a small number (29%, n = 4) provided education to select families (≤50% of families). Families that were targeted for education included those with a child that had fat- and protein-related glycaemic excursions (100%, n = 14), those that asked (93%, n = 13), and families with a child who ate high-fat and high-protein foods (86%, n = 12).

Almost all respondents (95%, n = 19) routinely used CGM to assess postprandial control, and 84% (n = 16) reported they were moderately to very confident in identifying fat- and protein-related glycaemic excursions. The CGM trace was used by most respondents (85%, n = 17) as a visual tool to educate families on the glycaemic impact of fat and protein. To teach families to identify foods that contained fat and/or protein, half of all respondents (50%, n = 10) used food groups (i.e., meat, dairy), 35% (n = 7) used ranges of fat and protein content (i.e., low, medium, and high), and 15% (n = 3) used thresholds (i.e., count if >20 g fat or protein).

All respondents (n = 20, 100%) indicated that they would like training and resources for themselves and resources for families to support fat and protein education. The majority (55%, n = 11) preferred a combination of online, self-paced, and live web-based delivery, followed by online, self-paced, and in-person (25%, n = 5) and online, self-paced only (20%, n = 4).

### 3.4. Fat and Protein Insulin Management Strategies

Of the 50% of respondents (n = 10) who did not always recommend strategies to manage fat and protein where there was evidence of an impact, the majority (90%, n = 9) reported ‘often’ recommending strategies, while one reported ‘sometimes’ recommending strategies.

The types of insulin strategies used by respondents to manage the glycaemic impact of fat and protein varied across the modes of insulin delivery, including (i) multiple daily injection (MDI) therapy, (ii) open-loop insulin pump therapy (IPT), and (iii) AID systems. In MDI therapy, most respondents recommended splitting the meal insulin dose (85%, n = 17) and increasing the insulin dose (75%, n = 15). In open-loop IPT, the most common strategies were giving meal insulin in an extended bolus (90%, n = 18), where a portion of the insulin dose is delivered prior to a meal and the remaining portion delivered at a constant rate over a set time-period, and increasing the insulin dose (80%, n = 16).

Sixty percent (n = 12) of respondents recommended strategies in AID systems. The strategies used were splitting the meal carbohydrate entry (80%, n = 12), entering additional ‘fake’ carbohydrates after the meal (10%, n = 2), entering additional ‘fake’ carbohydrates in addition to what the child is planning to eat, and using an advanced pump dosing feature such as ‘boost’ (increases the amount of insulin delivered as autocorrections) and extended bolus (delivers meal insulin over a set period of time) (n = 2, 10%). Forty percent of respondents (n = 8) reported letting ‘the autocorrections take care of it’, i.e., allowing the pump to attempt to maintain euglycaemia via the delivery of correction insulin doses without input from the user.

Most respondents, (90%, n = 18) used the CGM trace to guide adjustments to the insulin strategy, and 80% (n = 16) reported that they felt moderately to very confident in doing so. However, of the latter group, almost half (44%, n = 7) did not have a target for the peak postprandial glucose rise, and a quarter (23%, n = 4) did not have a target for when the postprandial glucose level should return to baseline.

### 3.5. Barriers to the Provision of Education on the Glycaemic Impact of Fat and Protein

All respondents (100%, n = 20) believed that providing education on the glycaemic impact of fat and protein was their (the dietitian’s) role, yet over two-thirds (70%, n = 14) did not provide education on the glycaemic impact of fat and protein to all families at their centre. Of all respondents, 75% (n = 15) reported that there were a number of barriers to the implementation of fat and protein education; these included a lack of (i) procedure/policy on implementation of education (46.7%, n = 7), (ii) resources for families (40%, n = 6), (iii) time (33.3%, n = 5), and (iv) perceived interest from the family (33.3%, n = 5). Short responses in this section also found respondents (20%, n = 3) believed some families did not have sufficient health literacy to understand the glycaemic impact of fat and protein. For example, ‘*Protein and fat are not taught if it is not appropriate for a family who is struggling with carb counting or when health literacy is low*’, or ‘*families do not have the capacity*’.

### 3.6. Barriers to the Provision of Insulin Strategies to Manage the Glycaemic Impact of Fat and Protein

Half of all respondents (50%, n = 10) did not always provide strategies to manage fat and protein where there was evidence of an impact on postprandial glycaemia. The primary reasons given for not providing strategies were (*i*) they perceived the family was already overwhelmed (100%, n = 10), (*ii*) they believed the child and/or family lacked interest (60%, n = 6), and (*iii*) they were unsure of the best strategy to use (40%, n = 4). Only one respondent (5%) identified that they were concerned about hypoglycaemia and that they may be perceived to be promoting ‘unhealthy’ foods.

### 3.7. Barriers to the Use of CGM

CGM was identified as a key tool used by respondents to educate families on the glycaemic impact of fat and protein (85%, n = 17) and to guide adjustments to the insulin strategy for fat and protein foods (90%, n = 18). Of respondents, 65% (n = 13) reported barriers to using the CGM trace to identify fat- and protein-related glycaemic excursions and guide adjustments to the insulin strategy. These barriers included time in clinic appointments (85%, n = 11), insufficient CGM and dietary data (38%, n = 5), a lack of procedure on how to interpret the CGM and what actions to take (31%, n = 4), and an inability to access CGM data (31%, n = 4). It should be noted that in New Zealand, CGM is not subsidized by the government, whereas in Australia, CGM is fully subsidized. Only 17% (n = 1/6) of New Zealand respondents reported CGM was used by >75–100% of their centre’s population, compared to 64% (n = 9/14) of Australian respondents. Of New Zealand respondents, 67% (n = 4/6) uniquely identified the affordability of CGM as a barrier to use.

## 4. Discussion

This is the first study to investigate the implementation of best-practice, international nutrition guidelines on the glycaemic management of dietary fat and protein in paediatric T1D care. This is a relatively new area in T1D nutritional management where carbohydrate counting has historically been the core practice. It was found that of 20 paediatric T1D dietitians at specialist T1D tertiary centres across Australia and New Zealand, only 10% provided care that was fully consistent with guidelines. Most respondents did not educate all families on the impact of fat and protein, nor did they provide this education at the recommended timepoint, i.e., at diagnosis. Further, only half of all dietitians always provided families with insulin strategies to manage fat and protein when there was evidence of an impact on the child/adolescents’ glucose levels. Dietitians identified several key barriers to the provision of guideline-recommended care. Organizational barriers such as an absence of centre-based policy and resources for families hampered education. Individual barriers, such as the belief that families were already overwhelmed or not interested and knowledge around evidence-based methods, prevented dietitians from providing strategies.

There are no evidence-based approaches to support the implementation of international T1D nutrition guidelines, leading to significant variability in practice worldwide [18]. In the present study, the timing of fat and protein education delivery and methods of teaching varied considerably, with six different timepoints for delivery and three different methods of teaching fat and protein quantification identified. Guidelines identify diagnosis as the preferred time for education delivery [6], as this is a pivotal time for the development of foundational nutrition knowledge and diabetes self-care behaviours that will persist throughout the lifespan. Research has not yet defined how accurately fat and protein need to be counted to optimise postprandial glycaemia, and to that point, the superiority of one method over another. However, success in fat and protein insulin dosing ultimately requires some fat and protein awareness to minimise errors in dosing and enable ongoing dose adjustment. Previous research has identified high-fat (>20 g), high-protein (>30 g), and low-fat (<10 g) foods as targets for education [19].

In addition to providing education on the impact of fat and protein on glycaemia, guidelines emphasise the importance of adjusting the insulin dose and timing of delivery to compensate for their delayed and prolonged impact [6,12]. Consistent with guidelines [6], in those using open-loop IPT and MDI therapy, most dietitians (85–90%) recommended increasing the insulin dose and delivering it in an extended bolus or as a split dose, respectively. There are currently no guideline-recommended strategies for insulin dosing for fat and protein in those using AID systems. In the absence of such recommendations, 60% of dietitians routinely gave advice to increase the meal insulin dose via the entry of additional ‘fake’ carbohydrates without scientific evidence. A growing body of research supports the clinical experience of a continued need for adjustments for dietary factors other than carbohydrate in AID users [20]. Tornese et al. demonstrated a significant ~2.2 mmol/L increase in mean 2-h postprandial glucose with the addition of 20 g of fat to 30 g of simple carbohydrate [21]. Further controlled studies are required to identify the most effective insulin strategies for optimising glycaemia following high-fat and high-protein meals in AID system users and serve as an evidence base for guideline recommendations.

CGM data reports provide users and clinicians with novel insights into the impact of different foods and insulin dosing behaviours on the dynamics of the postprandial glucose profile [13] and a host of metrics to measure and describe this impact [22]. In the present study, almost all dietitians (85–90%) reported using the CGM trace and glycaemic metrics to educate families on the glycaemic impact of fat and protein and to guide adjustments to the insulin strategy. While most reported confidence in the latter, many did not have clearly defined targets for the peak postprandial glucose level and when the glucose level should return to baseline after eating. Further research is required to establish such targets.

The present study identified several key barriers to guideline implementation around the impact of fat and protein on glycaemia and management strategies. Almost half of all dietitians identified a lack of centre-based policy and consumer resources as the primary barriers to the provision of education. In relation to both the provision of education and management strategies, dietitians expressed concern that the concepts were too complex for some families and presented an additional, unnecessary burden. The health-care professional team, specifically the dietitian, has previously been identified by families as the primary source of information regarding insulin dosing strategies [13]. Hence, dietitians must be equipped with the knowledge and resources and actively provide families with this evidence-based information or risk families seeking information elsewhere and adopting potentially harmful strategies such as food avoidance or restriction [13]. As per international consensus guidelines, educational programs (and associated resources) should take a person-centred approach and be adaptable to the age, stage of diabetes, maturity, lifestyle, culture, and learning pace of the child and adolescent [23]. Further, training dietitians in teaching principles and practice is needed to enable successful implementation of evidence-based educational programs [23].

This study has strengths and limitations. The primary strength of the study was the 100% response rate, with all participants responding to 100% of the questions in the questionnaire. This provides a representative understanding of education practices across tertiary clinics in Australia and New Zealand. To minimise social desirability bias, the survey was conducted anonymously. Limitations of the study were that the survey was not validated; however, no validated survey exists, and the survey was developed and piloted with specialist input and informed by clinically relevant challenges noted at T1D national meetings. Additionally, the study was conducted on a subset of dietitians working at specialist paediatric tertiary centres (>100 children and adolescents with T1D) in a discrete geographical region. While selection of tertiary centres only was intentional in design, as they generally provide outreach education to rural and remote centres, this potentially limits generalisability to those working in rural and remote locations, as well as internationally. The use of descriptive statistics also limits generalisability. Future studies could include the perspective of a larger number of dietitians working in these settings. A larger sample of dietitians would aid in better understanding the relationships between adherence to guidelines and other variables such as years of experience.

## 5. Conclusions

Guidelines are statements systematically designed to provide clinicians with the best available evidence to support clinical decision-making and deliver care that optimises health outcomes. This study has shown that implementation of international nutrition guidelines on the glycaemic management of dietary fat and protein in paediatric T1D is challenging and inconsistent. Ninety percent of dietitians surveyed were not providing fat and protein education and management strategies in line with guidelines. Study findings have defined a central role for CGM in both the education of families on the glycaemic impact of fat and protein and in insulin dose adjustment for fat and protein. To enable guideline implementation, dietitians need greater support through education resources for families and evidence-based strategies for implementation that are outlined in centre-based policies and procedures.

## Figures and Tables

**Figure 1 nutrients-17-01109-f001:**
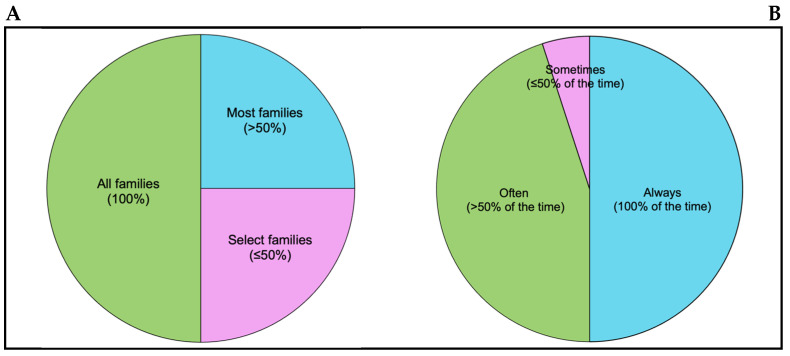
(**A**) The proportion of families that dietitians educate on the glycaemic impact of fat and protein. (**B**) The frequency at which dietitians provide strategies to manage fat and protein where there is evidence of an impact on glycaemia.

**Figure 2 nutrients-17-01109-f002:**
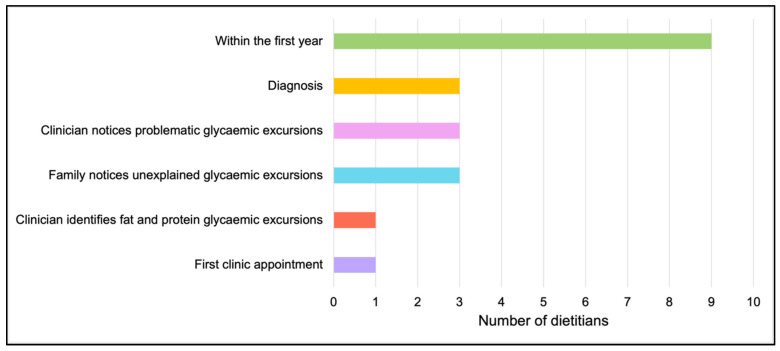
Timing of education delivered on the impact of fat and protein on postprandial glycaemia.

**Table 1 nutrients-17-01109-t001:** Demographic characteristics of survey respondents (n = 20).

Characteristic	Number of Respondents	Proportion of Respondents (%)
**Location of workplace**		
** *Australia* **	**14**	**70**
Australian Capital Territory	1	5
New South Wales	6	30
Northern Territory	1	5
Queensland	2	10
South Australia	1	5
Tasmania	1	5
Victoria	1	5
Western Australia	1	5
** *New Zealand* **	**6**	**30**
**No of children with T1D in care**		
0–200	11	55
201–400	4	20
401–600	1	5
601–1000	2	10
1000+	2	10
**Years of experience in T1D care**		
0–5	4	20
6–10	8	40
11–15	3	15
16–20	4	20
21–25	1	5
**Av number of reviews per child annually**		
1	9	45
2	8	40
3	2	10
4	1	5

## Data Availability

The original contributions presented in this study are included in the article/Appendix A. Further inquiries can be directed to the corresponding author(s).

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
