# Peer review of "Education Practices of Dietitians Across Australia and New Zealand Around the Glycaemic Management of Dietary Fat and Protein in Type 1 Diabetes and the Use of Continuous Glucose Monitoring: A Survey Evaluation"

_nutrients, 2025, doi:10.3390/nu17071109_

Round 1

Reviewer 1 Report

Comments and Suggestions for Authors

Article review: Education practices of dietitians across Australia and New Zealand around the glycemic management of dietary fat and protein in type 1 diabetes and the use of continuous glucose monitoring: A survey evaluation

Dear Authors 
Thank you for the opportunity to review this article, which presents the results of a survey evaluation of the educational practices of pediatric dietitians in Australia and New Zealand around the glycemic management of dietary fat and protein in type 1 diabetes (T1D). The issues addressed in the paper are of significant practical importance, as educating patients with T1D is already challenging with respect to carbohydrates, and incorporating fat and protein issues requires an even more complex approach and additional resources. 
The study provides valuable data on the application of international guidelines, which state that patients should receive education on the effects of fats and proteins on glycemia from the moment of diagnosis, in actual clinical practice and the existing barriers to implementation.
The work was well planned and carried out according to methodological standards, allowing other researchers to replicate the procedure.
The results are presented in a consistent and clear manner. In the discussion, the authors have included an apt analysis of the limitations that can be addressed in future research.
In conclusion, the article makes an important contribution to diabetes education research and highlights the need for further support for dietitians in effective glycemic management strategies. 
I do not find any significant aspects in need of improvement.

Author Response

REVIEWER 1 

No comments to address. Thank you kindly for your review of, and support for the manuscript. 

Reviewer 2 Report

Comments and Suggestions for Authors

This research underscores the pivotal role of education in the effective management of type 1 diabetes. 

I suggest reading and including this consensus. 

DOI: 10.1111/pedi.13418

Some errors related to text editing, like: centres (line 26) or The primary aim of this study were should be corrected. 

Author Response

REVIEWER 2

This manuscript presents a survey study evaluating how dietitians in Australia and New Zealand educate families of children with type 1 diabetes (T1D) on the glycaemic impact of dietary fat and protein and the extent to which they use continuous glucose monitoring (CGM) to guide insulin strategies. The study identifies gaps between clinical guidelines and current practices highlighting barriers such as lack of resources, unclear policies, and perceived family burden.

Comments:

  1. Only 20 dietitians were surveyed, representing tertiary centres. While this ensures expertise, it limits generalizability to community settings. Acknowledge the limitations of a tertiary-care-only sample and suggest future studies including primary care dietitians.

Response 1:

Thank you for your comment. Lines 349-51 (page 9) of the manuscript acknowledge a potential lack of generalisability to rural and remote centres and those in other geographical locations as a limitation of the study-

Additionally, the study was conducted in a subset of dietitians working at specialist paediatric tertiary centres (>100 children and adolescents with T1D) in a discreet geographical region. This potentially limits generalisability to those working in rural and remote locations, as well as internationally.’

We have further expanded on this comment to suggest that ‘Future studies could include the perspectives of dietitians working in these settings.’ Please see page 9, 354-55 for changes.

  1. The study does not include endocrinologists or diabetes educators who also influence T1D dietary education. Include perspectives from endocrinologists or caregivers to provide a fuller picture of T1D education.

Response 2:

In the multi-disciplinary health team dietitians are the primary person responsible for delivering nutrition education. International clinical guidelines identify the specialist paediatric diabetes dietitian as the person to provide medical nutrition therapy from diagnosis to children living with type 1 diabetes (https://onlinelibrary.wiley.com/doi/10.1111/pedi.13429) Further, an international survey study reported that at more than 80% of centres (53 included) nutrition education at diagnosis was delivered by the dietitian (ttps://onlinelibrary.wiley.com/doi/10.1111/pedi.13161).

We did not include the perspectives of caregivers and endocrinologists in this particular study as.

  • The present study was directly informed by and serves as a follow-up to a previous study by our group that documented the perspectives of families/ caregivers around the management of problematic foods including those high in fat and protein (https://onlinelibrary.wiley.com/doi/10.1111/1747-0080.12630). This has been previously noted in the introduction lines 67-70.
  • At tertiary centres the dietitian, endocrinologist (and diabetes educator) work as part of multi-disciplinary team, therefore the dietitian views are likely to reflect the views of the team. Involving other team members from the same site would have contributed to a duplication of responses.

Future research in the area could focus on surveying/ interviewing a wider range of health professionals in the diabetes team.

  1. The authors note that no validated survey exists, but additional validation steps (e.g., test-retest reliability) could enhance credibility. Consider validating the survey tool.

Response 3:

Thank you for your comment. This survey was specifically designed for and pilot tested by health care professionals working in specialist paediatric diabetes tertiary centres in Australia and New Zealand following standard, evidence-based survey design methodology and reporting standards and deemed to have reliable face validity. It is not the intention of the authors to broadly apply the survey to other populations however, the authors welcome the adaptation of the survey for other dietitian populations or health professionals as mentioned above.

  1. Including family experiences could provide a more comprehensive understanding.

Response 4:

The present study, which provides the perspective of the dietitian was directly informed by and serves as a follow-up to a previous study by our group that documented the perspectives of families/ caregivers around the management of problematic foods including those high in fat and protein (https://onlinelibrary.wiley.com/doi/10.1111/1747-0080.12630). A future planned study will examine the efficacy of an intervention to support guideline implementation. This study will document the perspectives of both the dietitian in implementing the intervention and the experience of the family/ caregiver and person living with type 1 diabetes in receiving the intervention.

  1. The study relies primarily on descriptive statistics, with only Fisher’s Exact Test applied for categorical associations.

Response 5:

Agree.

  1. Given the small sample size, logistic regression could have explored relationships between years of experience and adherence to guidelines.

Response 6:

Thank you for your suggestion. As noted, this study has a small sample size. Given this, a logistic regression would be underpowered to detect true relationships between the years of experience and adherence to guidelines. We have added this to the manuscript as an avenue for future research, ‘A larger sample of dietitians are needed to better understand the relationships between adherence to guidelines and other variables such as years of experience’. Please see page 9, lines 355-57.

  1. Do rural vs. metropolitan dietitians differ in their approaches? Expand discussion on how current practices compare to international surveys of dietitians in other regions.

Response 7:

In Australian and New Zealand outreach models’ dietitians working in tertiary centres provide practice updates, education and mentoring to dietitians working in regional and rural settings ensuring relative consistency in care delivery.

There are currently no studies which explore the education practices of dietitians around the management of fat and protein in type 1 diabetes. There is one published international survey study that explores nutrition education practices. The study design varied significantly. The sample population was primarily endocrinologists with a focus on carbohydrate counting and meal routines.

  1. Conduct subgroup analyses to explore whether experience, location, or clinic size affects adherence to guidelines.

Thank you for your suggestion. While we are not powered to perform sub analyses, we agree that this would be of interest and will be considered in power calculations for any future studies involving rural and remote centres.

  1. Discuss how education gaps might impact patients using different insulin delivery methods.

Response 9:

Currently, international best practice guidelines around the management of dietary fat and protein are based on studies conducted in those using multiple daily injection and open-loop insulin pumps (non-automated) therapies. There are no guidelines which specify insulin dosing strategies for users of newer automated insulin delivery (AID) systems. There is a paucity of studies in this population to inform guidelines specifically for AID systems users.

To acknowledge the lack of guideline recommended strategies for AID system users the following statement has been included in the discussion section of the manuscript, ‘There are currently no guideline recommended strategies for insulin dosing for fat and protein in those using AID systems.’ (see page 8, lines 307-308).  We have also acknowledged that the impact of this is that dietitians are adopting strategies without evidence of its efficacy as per the following statement ‘In the absence of such recommendations, sixty percent of dietitians routinely gave advice to increase the meal insulin dose via the entry of additional ‘fake’ carbohydrates without scientific evidence.’ (see page 8, lines 308-310).

  1. Proofread for minor grammatical errors to enhance readability.

Response 10:

The manuscript has been proofread and grammatical errors corrected.

Reviewer 3 Report

Comments and Suggestions for Authors

This manuscript presents a survey study evaluating how dietitians in Australia and New Zealand educate families of children with type 1 diabetes (T1D) on the glycemic impact of dietary fat and protein and the extent to which they use continuous glucose monitoring (CGM) to guide insulin strategies. The study identifies gaps between clinical guidelines and current practices highlighting barriers such as lack of resources, unclear policies, and perceived family burden.

Comments:

  • Only 20 dietitians were surveyed, representing tertiary centers. While this ensures expertise, it limits generalizability to community settings. Acknowledge the limitations of a tertiary-care-only sample and suggest future studies including primary care dietitians.
  • The study does not include endocrinologists or diabetes educators who also influence T1D dietary education. Include perspectives from endocrinologists or caregivers to provide a fuller picture of T1D education.
  • The authors note that no validated survey exists, but additional validation steps (e.g., test-retest reliability) could enhance credibility. Consider validating the survey tool.
  • Including family experiences could provide a more comprehensive understanding.
  • The study relies primarily on descriptive statistics, with only Fisher’s Exact Test applied for categorical associations.
  • Given the small sample size, logistic regression could have explored relationships between years of experience and adherence to guidelines.
  • Do rural vs. metropolitan dietitians differ in their approaches? Expand discussion on how current practices compare to international surveys of dietitians in other regions.
  • Conduct subgroup analyses to explore whether experience, location, or clinic size affects adherence to guidelines.
  • Discuss how education gaps might impact patients using different insulin delivery methods.
  • Proofread for minor grammatical errors to enhance readability.

Overall, manuscript is well-structured and presents valuable insights but requires improvements to strengthen its impact.

Author Response

No reviewer 3. 

Round 2

Reviewer 3 Report

Comments and Suggestions for Authors

I appreciate authors for considering and addressing all my comments appropriately.

I have no further comments on this manuscript.